

# Navigating the amino acid sequence space between functional proteins using a deep learning framework

Tristan Bitard-Feildel[1,2]

[1] IBPS, CNRS, Laboratoire de Biologie Computationnelle et Quantitative, Sorbonne Université, Paris, France
[2] Institut des Sciences du Calcul et de des Données (ISCD), Sorbonne Université, Paris, France

## ABSTRACT

**Motivation:** Shedding light on the relationships between protein sequences and functions is a challenging task with many implications in protein evolution, diseases understanding, and protein design. The protein sequence space mapping to specific functions is however hard to comprehend due to its complexity. Generative models help to decipher complex systems thanks to their abilities to learn and recreate data specificity. Applied to proteins, they can capture the sequence patterns associated with functions and point out important relationships between sequence positions. By learning these dependencies between sequences and functions, they can ultimately be used to generate new sequences and navigate through uncharted area of molecular evolution.

**Results:** This study presents an Adversarial Auto-Encoder (AAE) approached, an unsupervised generative model, to generate new protein sequences. AAEs are tested on three protein families known for their multiple functions the sulfatase, the HUP and the TPP families. Clustering results on the encoded sequences from the latent space computed by AAEs display high level of homogeneity regarding the protein sequence functions. The study also reports and analyzes for the first time two sampling strategies based on latent space interpolation and latent space arithmetic to generate intermediate protein sequences sharing sequential properties of original sequences linked to known functional properties issued from different families and functions. Generated sequences by interpolation between latent space data points demonstrate the ability of the AAE to generalize and produce meaningful biological sequences from an evolutionary uncharted area of the biological sequence space. Finally, 3D structure models computed by comparative modelling using generated sequences and templates of different sub-families point out to the ability of the latent space arithmetic to successfully transfer protein sequence properties linked to function between different sub-families. All in all this study confirms the ability of deep learning frameworks to model biological complexity and bring new tools to explore amino acid sequence and functional spaces.

Corresponding author
Tristan Bitard-Feildel,
tristan@bitardfeildel.fr

# INTRODUCTION

Protein diversity, regarding sequence, structure, or function, is the result of a long evolutionary process. Protein fitness and natural selection lead to the current observation

of only a fraction of all possible amino acid sequence combinations and therefore, structures, and functions (*Dryden, Thomson & White, 2008*). These observed sequences are also referred to the amino acid sequence space. The sequence space is difficult to computationally explore due to its huge size and thus the constrains between sequence positions are hard to understand (*Axe, 2004*; *Luisi, Chiarabelli & Stano, 2006*; *Dryden, Thomson & White, 2008*; *Marchi et al., 2019*). The classification of amino acid sequences into protein domain families allows to organize the sequence space and reduce its complexity.

Many resources have been developed sover the years to group amino acid sequences into families whose members share sequence and structural similarities (*Dawson et al., 2016*; *Pandurangan et al., 2018*; *El-Gebali et al., 2018*). Thus, protein families permit to organize the sequence space. The sequence space area between these families is however mostly uncharted (*Das, Dawson & Orengo, 2015*) in spite of very remote evolutionary relationships between families (*Alva et al., 2010*). Navigating the sequence space with respect to the functional diversity of a family is therefore a difficult task. This difficulty is even increased by the low number of proteins with experimentally confirmed function. In this regard, computer models are needed to explore the relationships between sequence space and functional space of the protein families (*Goldstein & Pollock, 2016*; *Tian et al., 2018*; *Copp et al., 2018*; *Salinas & Ranganathan, 2018*; *Tubiana, Cocco & Monasson, 2019*; *Poelwijk, Socolich & Ranganathan, 2019*; *Russ et al., 2020*). Perfect modeling of the sequence space could have applications in molecular engineering, functional annotation, or evolutionary biology. It may for example be possible to understand completely the relationships between amino acid positions of a family responsible of a molecular function or to navigate the sequence space between families of different functions.

In this study, tools and strategies based on an unsupervised deep learning approach are proposed to model and navigate the current evolutionary uncharted area of the amino acid sequence space.

Previous deep learning generative models such as variational autoencoders (VAE) have been applied on biological and chemical data. They have for example been used to explore and classify gene expression in single-cell transcriptomics data (*Lopez et al., 2018*), or to explore the chemical space of small molecules for drug discovery and design (*Rampasek et al., 2017*; *Gómez-Bombarelli et al., 2018*). Their ability to reduce input data complexity in a latent space and perform inference on this reduced representation make them highly suitable to model, in an unsupervised manner, complex systems. Regarding protein science, VAE have been able to accurately model amino acid sequence and functional spaces (*Sinai et al., 2017*), to predict mutational impact (*Hopf et al., 2017*; *Riesselman, Ingraham & Marks, 2018*), to decipher protein evolution and fitness landscape (*Ding, Zou & Brooks, 2019*) or to design new proteins (*Greener, Moffat & Jones, 2018*). In this study, Adversarial AutoEncoder (AAE) network (*Makhzani et al., 2015*) is proposed as a new and efficient way to represent and navigate the functional space of a protein family. Unlike VAE, AAE networks constrain the latent space over a prior distribution. Prior distribution allows better inference to explore the whole latent distribution which is particularly useful for travelling through uncharted area (*Kadurin et al., 2017*).

Like VAE and other autoencoder architectures, AAEs reduce high dimensional data by projection, using an encoder, into a lower dimensional space. This space is known as a latent space, or embedding representation. The latent space can, in turn, be used by the decoder to reconstruct the initial data. AAE (*Makhzani et al., 2015*) architecture corresponds to a probabilistic autoencoder but with a constraint on the latent space of the encoder. The latent space is constrained to follow a defined prior distribution. This constraint is applied using a generative adversarial network (GAN) (*Goodfellow et al., 2014*) trained to discriminate between the latent space and the prior distribution. It ensures that meaningful samples can be generated from anywhere in the latent space defined by the prior distribution. Applied to biological sequences of a protein domain family, it is then possible to encode the sequence diversity to any prior distribution. Thus, the model is able to sample and generate new amino acid sequences of the family from any point of the prior distribution. Ideally, the learned latent space should be representative of the functions of the protein domain family and even able to dissociate protein sequences with different sub-functions.

Protein sequences can cluster in the latent space of the AAE network. These clusters were analyzed to verify their ability to group sequences according to function as observed with VAE networks. Three protein families including different sub-families were used to train AAE models. The protein functional annotations of these families were used to analyze the clustered sequences. The three different protein families selected were the sulfatases, the HUP (HIGH-signature proteins, UspA, and PP-ATPase) and the TPP (Thiamin diphosphate (ThDP)-binding fold, Pyr/PP domains) families. The sulfatases are a group of proteins acting on sulfated biomolecules. This family have been manually curated into sub-family with specific functions according to substrate specificity (*Barbeyron et al., 2016*). They are found in various protein family databases, such as in Pfam (PF00884). The SulfAtlas database (*Barbeyron et al., 2016*) is a collection of curated sulfatases centered on the classification of their substrate specificity. The majority of Sulfatases (30,726 over 35,090 Version 1.1 September 2017) is found in family S1 and is sub-divided into 73 sub-families corresponding to different substrate specificities. Sub-families S1-0 to S1-12 possess proteins with experimentally characterized EC identifiers.

The two other protein families, HUP and TPP families are not manually curated but were selected as they are known to have multiple functions (*Das, Dawson & Orengo, 2015*). Proteins of the HUP family are a very diverse group with functions linked to particular motifs such as HIGH and KMSKS (nucleotidyl transferases and t-RNA synthetases activities), ATP PyroPhosphatase motif, or sequence motifs responsible of the hydrolysis of the alpha-beta phosphate bond of ATP (*Bork & Koonin, 1994*; *Wolf et al., 1999*; *Aravind, Anantharaman & Koonin, 2002*). The TPP family is made of very similar protein domains which are probably evolutionary related (*Muller et al., 1993*; *Berthold et al., 2005*). They have pyruvate dehydrogenases, decarboxylate, and binding functions (*Muller et al., 1993*).

The VAE architecture has previously been used to cluster protein sequences and interpret the resulting clusters regarding their function or evolutionary history (*Sinai et al., 2017*; *Hopf et al., 2017*; *Riesselman, Ingraham & Marks, 2018*; *Ding, Zou & Brooks, 2019*; *Greener, Moffat & Jones, 2018*). These experiments have not studied the quality of the

architecture generative ability for protein sequences. In particular, the performances of the architecture is not known for the tasks of navigating the sequence space and transferring features between clusters. In this study, two experiments were carried out in this direction using latent space interpolation and latent space arithmetic operations. These experiments were designed as new tools and frameworks for the amino acid sequence space exploration.

Data point interpolations between protein sequences of different sulfatase sub-families was used to analyze the latent space coverage of the protein domain family functional space. The interpolated data points correspond therefore to unseen proteins, *i.e.* evolutionary uncharted area between groups of amino acid sequences. A good model should be able to produce realistic protein sequences from these data points.

This study also explored arithmetic operations with protein sequences encoded in their latent space to generate new protein sequences. Arithmetic operations on latent space have previously been reported to transfer features between images of different classes (*Radford, Metz & Chintala, 2015*). These operations may therefore have interesting potential for molecular design and for exploration of the amino acid sequence space. Four different strategies were explored to combine latent spaces of different sulfatase sub-families. The generated proteins from the combined latent spaces were analysed in term of sequences and structures, after being built by comparative modelling.

# METHODS

## Protein families

### The sulfatase family

An initial seed protein multiple sequence alignment (MSA) was computed from sequences of the protein structures of SulfAtlas (*Barbeyron et al., 2016*) database sub-families one to 12. This seed was used to search for homologous sequences on the UniRef90 (*Suzek et al., 2014*) protein sequence database using hmmsearch (*Eddy, 2011*) with reporting and inclusion e-values set at $1e^{-3}$.

A label was assigned to each retrieved protein if the protein belonged to one of the 12 known sub-families. The MSA computed with hmmsearch was filtered to remove columns and sequences with more than 90% and 75% gap characters respectively. Proteins with multiple hits on different parts of their sequences were also merged into a single entry. From 105181 initial protein sequences retrieved by hmmsearch, the filtering steps led to a final set of 41,901 proteins.

### HUP and TPP protein families

A similar protocol was followed for the HUP and TPP protein families. Instead of using an initial seed alignment made of sequences with known 3D structures, the CATH protein domain HMM (*Orengo et al., 1997*; *Sillitoe et al., 2018*) was used to search for homologous sequences in the UniRef90 database. CATH models 3.40.50.620 and 3.40.50.970 correspond to the HUP and TPP protein families, respectively. A sequence filtering pipeline identical to the one used for the sulfatase family was applied to each of the resulting MSAs.

The final numbers of proteins in each dataset were: 25041 for the HUP family (32,590 proteins before filtering) and 33,693 for the TPP family (13,3701 before filtering).

## Deep learning model

### Generative adversarial network

A complete description of Generative Adversarial Network can be found in *Goodfellow et al. (2014)*. To summarize, the GAN framework corresponds to a min-max adversarial game between two neural networks: a generator (G) and a discriminator (D). The discriminator computes the probability that an input $x$ corresponds to a real point in the data space rather than coming from a sampling of the generator. Concurrently, the generator maps samples $z$ from prior $p(z)$ to the data space with the objective to confuse the discriminator. This game between the generator and discriminator can be expressed as:

$$min_G \ max_D \ E_{x \sim p_{data}}[\log D(x)] + E_{z \sim p(z)}[log(1 - D(G(z)))] \tag{1}$$

### Adversarial auto-encoder

Adversarial autoencoders (AAEs) were introduced by (*Makhzani et al., 2015*). The proposed model was constructed using an encoder, a decoder networks, and a GAN network to match the posterior distribution of the encoded vector with an arbitrary prior distribution. Thus, the decoder of the AAE learns from the full space of the prior distribution. A Gaussian prior distribution was used in this study to compute the aggregated posterior $q(z|x)$ (the encoding distribution). The mean and variance of this distribution was learned by the encoder network: $z_i \sim N(\mu_i(x), \sigma_i(x))$. The re-parameterization trick introduced by (*Kingma & Welling, 2014*) was used for back-propagation through the encoder network.

Three different architectures were evaluated. The general architecture was as follows (see Table S1 and Fig. S1 for a representation of architecture number 3). The encoder was made of one or two 1D convolutional layers with 32 filters of size sevenand a stride of length two, and one or two densely connected layers of 256 or 512 units. The output of the last layer was passed through two stacked densely connected layers of hidden size units to evaluate $\mu$ and $\sigma$ of the re-parameterization trick (*Kingma & Welling, 2014*).

The decoder was made of two or three densely connected layers of the length of the sequence family time alphabet units for the last layers and of 256 or 512 units for the first or the two first layers. The final output of the decoder was reshaped to match the input shape. A softmax activation function was applied, corresponding to the amino acid probabilities at each position. To convert the probability matrix of the decoder into a sequence, a random sampling according to the probability output was performed at each position. The selected amino acid at a given position was therefore not necessarily the amino acid with the highest probability but reflect the biological distributions. The discriminator network was made of two or three densely connected layers. The last layer had only one unit and corresponds to the discriminator classification decision using a sigmoid activation function.

## Model training

The network was trained for each protein family independently. Amino acids and gap symbol of sequence input data were transformed using one-hot-encoding. A batch size of 32 was used to train the network. The autoencoder was trained using a categorical cross-entropy loss function between the input data and the predicted sequences by the autoencoder. The discriminator was trained using binary cross-entropy loss function between the input data encoded and the samples from the prior distribution.

## Generated sequences and structures analyses

### Dimensionality reduction

The AAE model can be used to reduce the dimensionality of the sequence space by setting a small latent size. Two dimensionality reductions were tested with latent size of two and 100. Latent size of two can be easily visualized and a larger latent size of 100 should represent the input data more efficiently as more information can be stored.

### Clustering

HDBSCAN (*Campello, Moulavi & Sander, 2013*; *McInnes & Healy, 2017*) was used to cluster the sequences in the latent space due to its capacity to handle clusters of different sizes and densities and its performances in high dimensional space. The Euclidean distance metric was used to compute distances between points of the latent space. A minimal cluster size of 60 was set to consider a group as a cluster as the number of protein sequences is rather large. The minimal number of samples in a neighborhood to consider a point as a core point was set to 15 to maintain relatively conservative clusters.

### Functional and taxonomic analyses

Enzyme functional annotation (EC ids) and NCBI taxonomic identifiers were extracted when available from the Gene Ontology Annotation portal (January 2019) using the UniProt-GOA mapping (*Huntley et al., 2014*). Proteins without annotation were discarded.

The annotation homogeneity was computed for each cluster. Considering a cluster, the number of different EC ids and taxonomic ids were retrieved. The percentage of each EC id (taxonomic id) was computed by cluster. An EC id (taxonomic id) of a cluster with a value of 90% indicates that 90% of the cluster members have this EC id (taxonomic id). A cluster with a high homogeneity value corresponds to functionally or evolutionary related sequences.

Homogeneous clusters will point out the ability of the AAE model to capture and distinguish protein sequences with functionally or evolutionary relevant features without supervision.

### Latent space interpolation

Twenty pairs of protein sequences were randomly chosen between all combinations of sulfatases sub-families with at least 100 labeled members but with less than 1,000 members (to avoid pronounced imbalance between classes): S1-0 (308 proteins), S1-2 (462 proteins), S1-3 (186 proteins), S1-7 (741 proteins), S1-8 (290 proteins) and S1-11 (669 proteins).

The coordinates of the selected sequences in the encoded latent space with 100 dimensions were retrieved. Spherical interpolations using 50 steps were performed between the pairs. Spherical interpolation has previously been reported to provide better interpolation for the generation of images (*White, 2016*). The interpolated points were given to the decoder to generate new sequences. Statistical analyses were carried out on the sequence transition from one family to an other. A model able to learn a generalized latent space should generate new sequences with smooth transitions between families. Analyses at the amino acid level were also performed on the interpolated sequences of two Sulfatase sub-families encoded far from one-another in the latent space.

### Shannon entropy computation

Shannon entropy is computed to measure the degree of variability at each position (column) of the MSA (*Jost, 2006*).

$$H(X) = -\sum_{i=1}^{n} P_i \log P_i \qquad (2)$$

with $P_i$ the frequency of symbol $i$ and $n$ the number of characters (20 amino acids and a gap symbol). The mean entropy per amino acid is computed for MSAs of biological sequences and generated sequences. Low entropy indicates that the analyzed sequences have low amino acid variability between each other. High entropy indicates high amino acid variability.

### Latent space arithmetic

It is possible to transfer features between data such as images by using subtraction or addition between projected data into a latent space (*Radford, Metz & Chintala, 2015*). This latent space property was tested on seven Sulfatase sub-families (S1-0, S1-1, S1-2, S1-3, S1-7, S1-8 and S1-11) selected on the basis of their number of protein sequences. Different arithmetic strategies (Fig. S2) were tested between latent spaces. The sub-family whose features are transferred is named the source sub-family. The sub-family receiving the transferred feature is named the query sub-family.

A first strategy consists in the addition of the mean latent space of the source sub-family to the encoded sequences of the the query sub-family. The second strategy differs from the first one by subtracting from the mean background latent space of all sub-families the latent space of the query sub-family. The third strategy differs from the second by the mean background strategy being computed using all sub-families except the source and query sub-families. Finally, in the fourth strategy, the subtraction is performed using a local KD-tree to only remove features shared by the closest members of a given query and the addition is performed by randomly selecting a member of the source family and its closest 10 members.

For each strategy, new sequences were generated using the latent spaces of all query proteins in the sub-families. The generated sequences by latent space arithmetic are compared to the initial query and source sub-families in terms of sequence and structural properties.

The protein sequence similarities were computed between the generated sequences by latent space arithmetic and the biological sequences of the two initial sub-families using a Blosum 62 substitution matrix. The sequence similarities were also computed inside a sub-family, between sub-families, and between generated sequences. The distributions of sequence similarities allow to explore the abilities of the latent space arithmetic operations and of the decoder to produce meaningful intermediate protein sequences from data points not corresponding to biological sequences. These data points correspond to an uncharted sequence space.

Protein structural models were computed using the structures of the initial sub-families as templates for MODELLER (*Webb & Sali, 2014*) and evaluated using the DOPE score (*Shen & Sali, 2006*). Models were computed using the generated sequences by latent space arithmetic on template structures from their source and query sub-families. The DOPE energies of the modeled structures were compared to structural models computed as references. The first structural model references were computed using the sequences and template structures belonging to the same sub-families, which should provide the best DOPE energies. The second structural model references were computed using the sequences and template structure belonging to different sub-families (ex: sequences from source sub-family and template structures from the query sub-family or inversely, sequences from query sub-family and template structures from the source sub-family), which should provide the worst DOPE energy. If the generated sequences by latent space arithmetic correspond to intermediate proteins with properties from two sub-families, they should have intermediate DOPE energies when compared to the others evaluated models.

## RESULTS

A structurally constrained MSA was computed using Expresso from T-Coffee webserver (*Armougom et al., 2006*; *Di Tommaso et al., 2011*) between sequences of S1 sulfatases structures. This MSA was processed into a Hidden Markov Model and hmmsearch was used to retrieve aligned sequence matches against the UniRef90 sequence database. A total of 76,427 protein sequence hits were found to match the sulfatase HMM in UniRef90. The sequences were filtered to remove columns and hits with more than 90% and 75%, respectively, of gap characters. The final MSA comprised 41,901 sequences. The sulfatases protein dataset was separated into a training, a validation, and a test sets with a split ratio of: 0.8, 0.1, and 0.1.

The three different AAE architectures (see "Method" section) were trained on the training set and evaluated on the validation set. The test set was only used on the final selected architecture. Models were evaluated by computing top k-accuracy, corresponding to the generation of the correct amino acid in the first $k$ amino acids. Table S2 shows the top k accuracy metric for k = 1 and k = 3 computed for the different AAEs. The accuracy scores scaled down with the number of parameters, but without any large difference. The architecture with the fewest number of parameters (architecture 3) was therefore selected to avoid over-fitting the data. The final accuracy scores on the test set were computed and were similar to the values observed during the model training: 62.5%

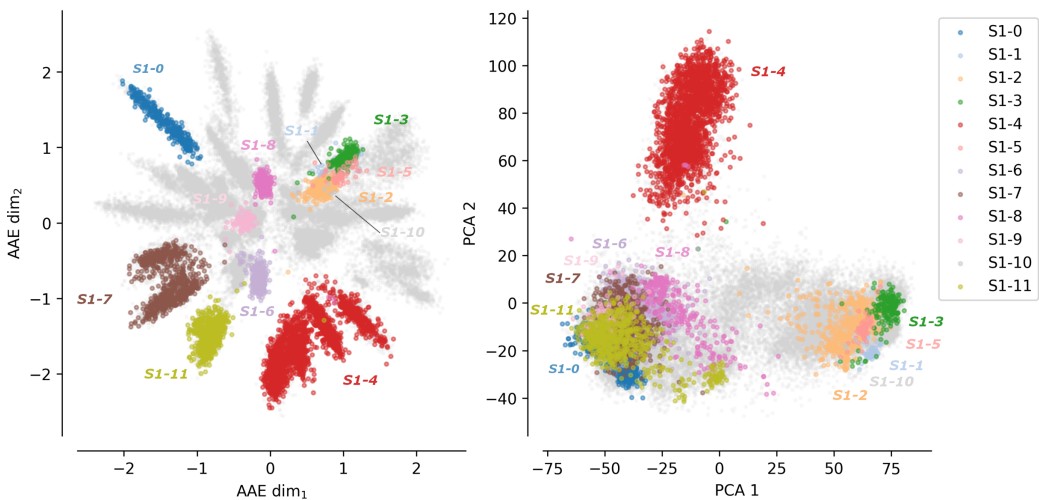

**Figure 1 Projections of the MSA sequences of the SulfAtlas family.** Left: projection in the encoding learned using an AAE (number of latent dimensions: 2). Right: projection using a PCA (two first components). Gray data points correspond to protein sequences not part of the curated 12 sub-families. This analysis is also performed for the HUP and TPP families. Results can be found in Fig. S3.

and 80.2% (k = 1 and k = 3). The selected architecture was separately trained using the protein sequences of the HUP and TPP families with identical train, validation, and test set splits.

## Latent space projection

AAE can be used as a dimensional reduction and visualization techniques by fixing the dimension of the latent space to two or three for plotting purpose. In this section, AAE network ability to create meaningful projection is tested on Sulfatase, HUP and TPP families by clustering and analysing protein sequences in terms of enzymatic activity and phylogenetic diversity.

Starting from the final MSA of the Sulfatase family, an AAE network was trained to project the sequences in a latent space with two dimensions. A PCA of the MSA was computed for comparison purpose with the AAE projection using the PCA first two principal components.

Figure 1 shows the protein sequences encoded by the AAE and the PCA projection. Each dot corresponds to a protein sequence. The dots are colored according to their sub-family. Gray dots correspond to protein sequences not belonging to any of the 12 curated sulfatases sub-families. The AAE displays in this figure a better disentanglement of the S1 family sequence and functional spaces than the PCA. Well-separated gray spikes can also be observed in the AAE projection. These spikes may correspond to groups of enzymes sharing common substrate specificity but not yet experimentally characterized.

In some cases, sub-families with identical functions are projected closely on the encoded space. For instance, sub-families S1-6 (light magenta) and S1-11 (yellow) have both the EC 3.1.6.14 activity (N-acetylglucosamine-6-sulfatase) and are closely located in the

encoded space. Moreover, some sub-family projections appear entangled such as the S1-1 sub-family (light blue, Cerebroside sulfatase activity, EC 3.1.6.8), the S1-2 (orange) and the S1-3 (green) sub-families (Steryl-sulfatase activity, EC 3.1.6.2), the S1-5 (pink) sub-family (N-acetylgalactosamine-6-sulfatase activity, EC 3.1.6.4), and the S1-10 (gray) sub-family (Glucosinolates sulfatase activity EC 3.1.6.-). The five families correspond to four different functions but are made of Eukaryotic protein sequences only and their entanglement may be due to their shared common evolutionary history. This separation based on the sequence kingdoms can clearly be visualized in the PCA projections with Eukaryotic sequences on the right side on sub-families with a majority of Bacteria sequences on the left side. The PCA projections failed to finely separate protein sub-families based on their functions. The example of protein B6QLZ0_PENMQ is also interesting. The protein is projected (yellow dot corresponding to the S1-11 sub-family) at coordinates (0.733,−1.289), inside the space of the S1-4 family (red). This may look like an error but a closer inspection shows that this protein is part of both the S1-4 and S1-11 sub-families of the SulfAtlas database.

Projections of sequences into latent spaces using AAE with two dimensions were also tested on the HUP and TPP families. The AAE projections can be visualized on Fig. S3. There are fewer functional annotations for these two families than for the sulfatase family. A strong separation can however clearly be observed between the major functions of the two families.

Latent spaces were evaluated for each protein family based on enzyme classification (EC) and taxonomic homogeneity. Given a set of protein sequences, the encoded sequences in a latent space of a 100 dimensions were clustered using HDBSCAN.

For the sulfatase family, 27 clusters were found, for which taxonomic and EC annotations could be extracted (Fig. S4 and Table S3). All these clusters displayed either strong taxonomic or EC homogeneity. Enzymatic homogeneity was higher than taxonomic homogeneity for 16 clusters, found equal in one cluster and lower for 10 clusters.

In the HUP family, all clusters had very high EC homogeneity (Table S4). Only two clusters out of 47 could be found with higher taxonomic homogeneity than EC homogeneity. For these two clusters enzymatic homogeneity values were high and only marginally different (cluster five, taxonomic homogeneity of 100% an EC homogeneity of 99% and cluster 31, taxonomic homogeneity of 99 % and EC homogeneity of 97%). Five clusters were found with equal taxonomic and EC homogeneity.

In the TPP family all clusters had also very high EC homogeneity (Table S5). Five clusters out of 51 could be found with higher taxonomic homogeneity than EC homogeneity. For these five clusters the differences between taxonomic homogeneity and EC homogeneity were higher than the differences observed for the HUP clusters. Six clusters were found with equal taxonomic and EC homogeneity.

The differences between the AAE and the PCA projections together with the general cluster enzymatic homogeneity highlight the ability of the encoding space to capture amino acid functional properties.

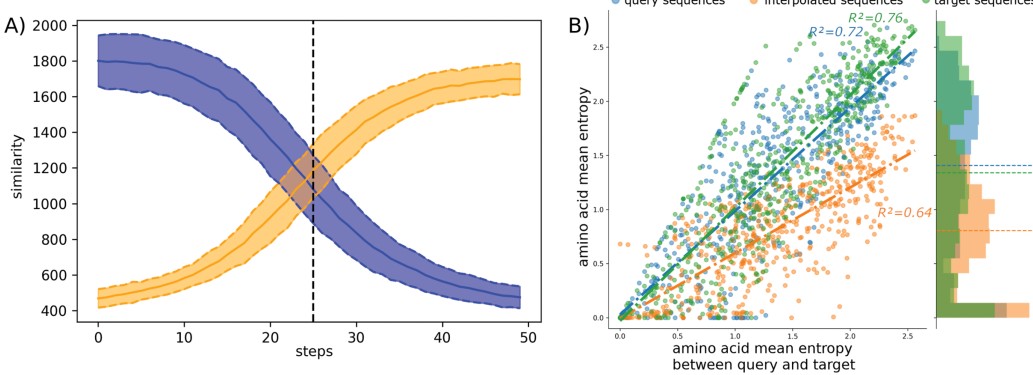

**Figure 2** **Interpolation analyses between sub-families S1-0 and S1-11.** (A) Sequence similarity distributions (sum of blosum weights, higher the score higher the similarity) between interpolated sequences and query proteins (blue) or target proteins (orange). (B) Distribution of amino acid Shannon entropy for interpolated sequences (orange, $R^2 = 0.64$) between sub-families S1-0 (blue, $R^2 = 0.72$) and S1-11 (green, $R^2 = 0.76$) over the amino acid mean Shannon entropy of query and target sub-families.

## Protein latent space interpolation

Interpolation between encoded sequences can be used to "navigate" between proteins of two sub-families. After the selection of a query sub-family, the sequences of this sub-family are projected to the latent space and used as the starting points of the interpolation. The end points of the interpolation correspond to sequences of a target sub-family, different from the query sub-family, and projected into the latent space. Twenty pairs of protein sequences were randomly selected between all combinations of protein sub-families to test the capacity of the encoded space and 50 intermediates, i.e. *interpolated*, data points were generated between each query/target pair. The sequence similarities were computed between the generated protein sequences from the interpolated latent space and the biological query and target protein sequences of the sub-families. It is thus possible to measure the amino acid sequence drift from one protein to another one.

The observed amino acid transitions from the query sub-family to the target sub-family are very smooth for all combinations of sub-families. The sequence similarity distributions display a logistic function shape as shown in Fig. 2A. The smooth transition between points demonstrates the ability of AAE network to encode the sequences into a smooth latent space and thus to correctly "fill" the gap between projected protein sequence sub-families.

The Shannon entropy was computed for each group of sequences: interpolated sequences between query and target sub-families, sequences of the query sub-families, and sequences of the target sub-families. Figure 2B shows the Shannon entropy distribution for the S1-0 and S1-11 sequences and their interpolated sequences. Interestingly, the figure shows lower entropy for interpolated sequences than for original sequences. Lower entropy indicates fewer amino acid variation at each position of the interpolated sequences than in biological sequences. Fewer amino acid variation at each position for the interpolated sequences could corresponds to restricted paths to travel between sub-families. This trend

is true for all interpolated sequences between all sub-families as reported in the Table S6. This is in agreement with molecular evolution theory and experiments that describe protein families as basins in fitness landscape (*Bornberg-Bauer & Chan, 1999*; *Sikosek, Chan & Bornberg-Bauer, 2012*; *Boucher et al., 2014*).

A closer inspection of interpolations between sub-families S1-0 and S1-4 (respectively blue and red data points in Fig. 1) was also performed to study changes at the amino acid level. The two sub-families are in "opposite" spaces in the two-dimensional projection. It can be observed in Fig. S5 that gapped area found in the query sequence but not in the target sequence (and inversely) are progressively filled (or broken down) starting from the flanking amino acids to the center of the gap (or inversely from the center to the flanking amino acids). This indicates an organized and progressive accumulation of amino acids (or gaps) that extend (or shrink) the region of the sequence previously without (or with) residues. For instance gap reduction can be observed in the generated sequences between sequence ID 2 of the sulfatase S1-0 family (query) and sequence ID 2196 of the sulfatase S1-4 family (target) at positions 75 to 86.

Moreover, family-specific amino acids are progressively replaced in key positions. In the previous interpolation between query and target sequences, it can thus be observed at positions 21 and 22 of the MSA a replacement of residues S and C by G and A (Fig. S5). Most transitions are not abrupt, and do not occur at the 25th-generated intermediate sequences but are smooth and correspond to plausible sequences. The ability of the AAE to generate interpolated sequences with emerging or disappearing features of two sub-families, highlights its capacity to generalize the decoding of latent space points not corresponding to encoded sequences and thus never observed during training, and outside the structured organization of the computed latent space.

## Protein latent space arithmetic

Latent space arithmetic is able to transfer learned features between different classes (*Radford, Metz & Chintala, 2015*). If applied to protein sequence latent space, this technique could permit to transfer features such as enzymatic activity or part of structure between protein families. To test this technique different arithmetic strategies (see "Methods" and Fig. S2) were tested between latent spaces of two Sulfatase sub-families. After performing the arithmetic operation between latent space coordinates, the protein sequences corresponding to the new coordinates were generated by the decoder. Protein structures of the generated sequences were computed using homology modeling The structure templates correspond to both sub-families. The protein structures of the generated sequences were compared to computed models using sequences and structures of the same sub-families and using sequences from one sub-family and structure templates from the other one. The Sulfatases sub-families S1-0, S1-2, S1-3, S1-7, S1-8 and S1-11 were chosen to test this technique.

In the following section, the terminology $\hat{S}$ eq. S1-XmY will correspond to a generated sequence using a combination of the mean latent space of the sub-family S1-Y added to the latent space of the sub-family S1-X. The X and Y sub-families will be referred to as the query and source sub-families.

First, two Prosite motifs of the Sulfatase family are analyzed from generated and original sequences. Figure S6 displays logo plots of two regions corresponding to Prosite motifs PS00523 and PS00149 to illustrate the amino acid content of the generated protein sequences by latent space arithmetic. These regions correspond to the most conserved regions of the sulfatase family and have been proposed as signature patterns for all the sulfatases in the Prosite database.

Different amino acid patterns can be observed between the sequence groups that can be classified as "competition", "taking over", or "balanced" pattern. A competition pattern of amino acids corresponds to equivalent frequency of two different amino acids in the generated sequences. A taking over pattern corresponds to an amino acid of one of the original sequences being the most frequent in the generated sequences. A balanced pattern corresponds to a maintained equilibrium between amino acids in the generated sequences. Some other positions are displaying much more complex patterns and cannot be summarized as a frequency competition between source and query sub-families. These behaviors can be observed several times through the logo plots but are still position-specific, meaning that the bits scores pattern observed in the source sub-families (Panels A and D of Fig. S6) do not necessary allow to predict the amino acids bits scores in the generated sequences (Panels B and C of Fig. S6).

Protein sequence similarities were computed to evaluate the diversity of the generated sequences and compare their diversity with the original sub-families. Protein sequence similarities were computed between: the generated sequences, the sequences of a sulfatase sub-family used to generate protein sequences, the generated sequences and their query sulfatase sub-family, the generated sequences and their source sulfatase sub-family, the query and source sequences of sulfatase sub-families. Figure 3 shows the mean and variance distribution of computed protein sequence similarities between these different groups for generated sequences computed using the first strategy. The first, second, and third strategies display a similar pattern and their corresponding figures are available in the Supplementary Information (Figs. S10, S12 and S14).

Protein sequence similarities between different sub-families (red upper triangles) have lower similarity scores and lower variances than the other distributions. Protein sequence similarities between sequences of a sub-family (blue circles) have the highest mean and variance values observed. However, since only 6 sub-families were kept for analysis (sub-families 0, 2, 3, 7, 8, and 11), trends must therefore be taken with precaution. Generated protein sequences compared to themselves (magenta lower triangles) have mean and variance protein sequence similarities higher than when compared to their query or sub-families. The last two (generated sequences compared to query sequences, orange squares and generated sequences compared to target sequences, green crosses) have mean and variance values spread between the blue and red distributions.

These distributions indicate that generated protein sequences by latent space arithmetic have an intrinsic diversity similar to the biological sub-families. Moreover, the generated sequences are less similar to the sequences from their query and source sub-families than to themselves. The generated sequences are also globally as similar to the sequences of their query sub-family as to the sequence of their source sub-family. The generation

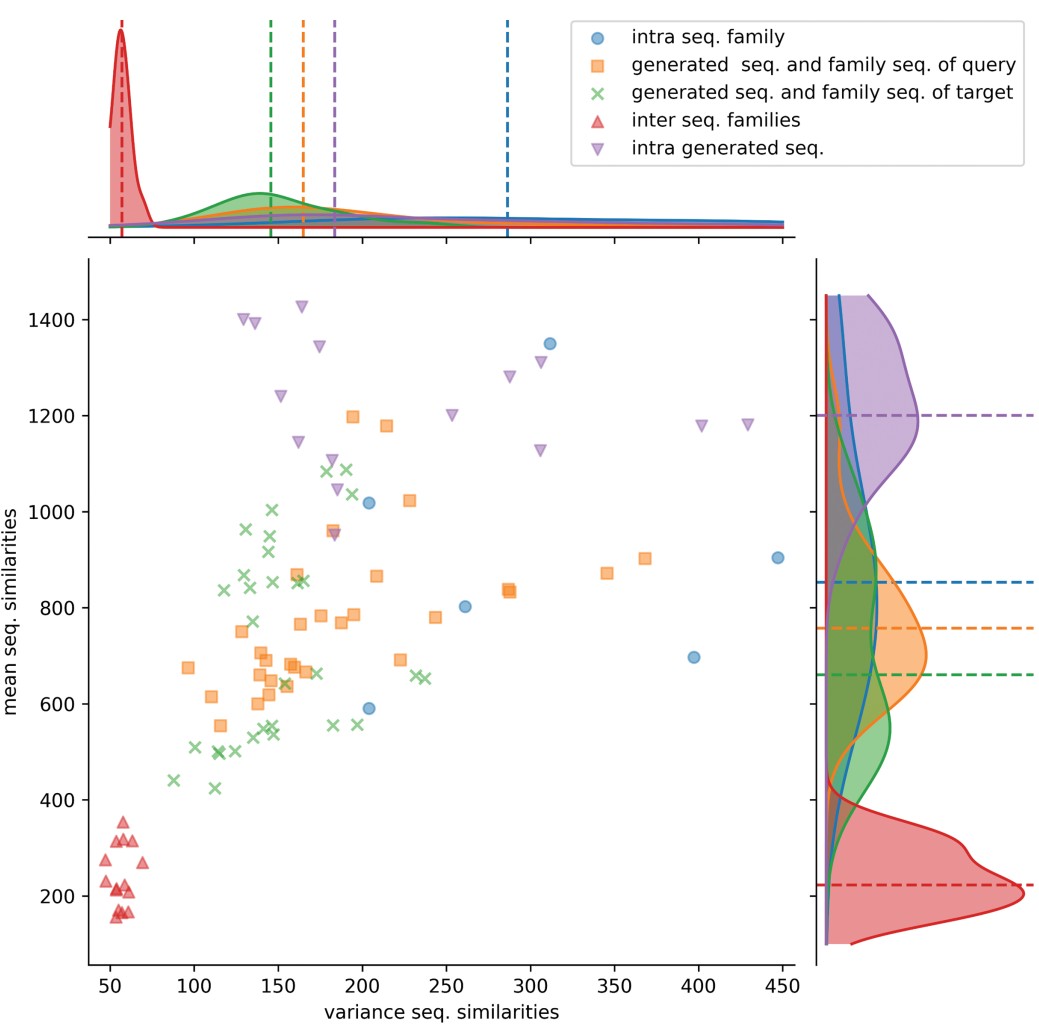

**Figure 3 Distributions of protein sequence similarities.** Blue dots: protein sequence similarity computed between sequences of the same protein sub-family. Orange squares: similarity computed between generated sequences and the sequences of their query sub-family (ex: S1-0m2 generated sequences and S1-0 sub-family sequences). Green x: similarity computed between generated sequences and the sequences of their target sub-family (ex: S1-0m2 generated sequences and S1-2 sub-family sequences). Red upper triangles: similarity computed between sequences of two different sub-families (ex: S1-0 sequences and S1-2 sequences). Magenta lower triangles: similarity computed between sequences of the same generated sequence group. The variance and the mean of each distribution are displayed on the horizontal and vertical axes.

process is therefore able to capture the features of the selected query and source sub-families and generate a protein sequence diversity similar to the original sub-families.

Finally, protein structure modeling was performed to assess and compare the properties of the generated sequences by latent space arithmetic and the protein sequences of the natural sub-families. For each sub-family, 100 original sequences were randomly selected along the corresponding generated sequences. All the generated sequences were aligned to protein structures of their corresponding source and query sub-families, and the alignments were used to create 3D structures models by comparative modeling. The quality of models was then evaluated with the DOPE function of MODELLER.

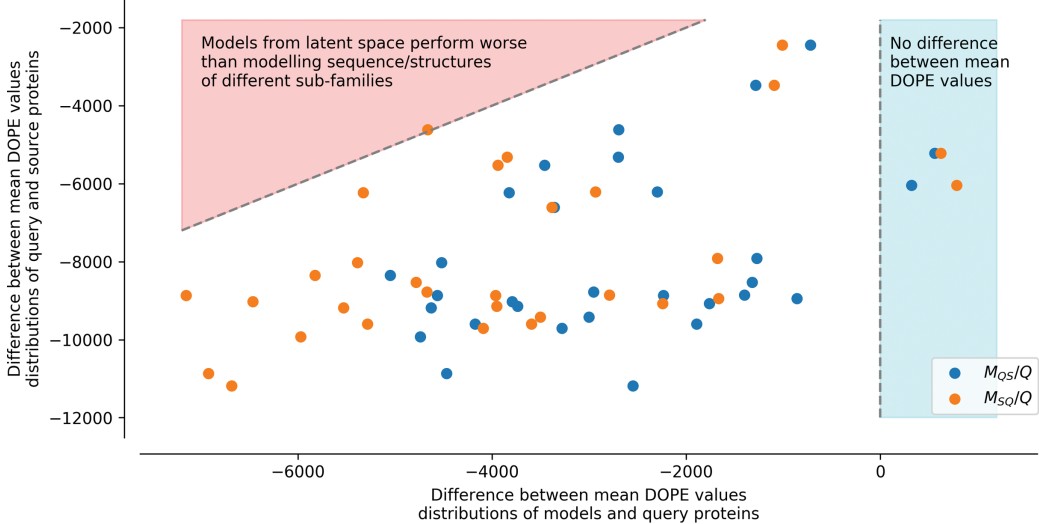

**Figure 4 Difference between mean DOPE distributions.** Mean value for each distribution, such as the distributions presented in Fig. S7, were computed. The *y* axis represents the difference between the mean values computed for query sequences modeled on structures of the same sub-family and mean values computed for source sequences modeled on structures of the query sub-family (ex: differences between mean of Struct. 0 Seq. 0 and mean of Struct. 0 Seq. 2 distributions in Fig. S7). The *x* axis corresponds to the difference between the mean values computed for query sequences modeled on structures of the same sub-family and mean values computed for query sequences to which latent spaces of the source sub-family sequences have been added and modeled on structures of the query sub-family (MQS/Q), or source sequences to which latent spaces of the query sub-family sequences have been added and modeled on structures of the source sub-family (MSQ/Q) (ex: differences between mean of Struct. 0 Seq. S1-0m2 and mean of Struct. 0 Seq. 0 distributions in Fig. S7). Points in the red area correspond to mean distribution values from generated sequences whose modeled structures have a higher energy than models created using pairs of sequences/structures from different sub-families. Points in the blue area correspond to mean distribution values from generated sequences whose modeled structures have a lower energy than models created using pairs of sequences/structures from the same sub-family.

Figure S7 shows an example of the energy distribution computed from models using the second strategy with query sub-family S1-0 and source sub-family S1-2. The lowest energies (best models) were found on modelled structures using the original protein sequences of a sub-family to the structural templates of the same sub-family (*Struct. 0 Seq. 0* and *Struct. 2 Seq. 2*). Conversely, the highest energies are found on modelled structures using the original protein sequences of a sub-family to the structural templates of another sub-family (*Struct. 0 Seq. 2* and *Struct. 2 Seq. 0*). Interestingly, generated sequences using additions and subtractions of latent spaces have intermediate energy distributions. This can be clearly observed in Fig. 4, where generated sequences are mostly situated between the two dotted lines. Dots on the right side of the vertical line at 0 correspond to modeled structures using sequences of the latent space with lower energy than the modeled structures using sequences from their original sub-family. Dots on the left side of the vertical line at 0 are modeled structures using sequences of the latent space with higher energy than the modeled structures using sequences from their original sub-family. The diagonal line on the top-left corner corresponds to the difference in energy between modeled structures using sequences from their original sub-family and modeled structures

using sequences from biological sequences of another sub-family. The energy of generated sequences modeled using their query sub-family templates (ex: *Struct. 0 Seq.S1-0m2* and *Struct. 2 Seq. S1-2m0* on Fig. S7 and $M_{QS}/Q$ on Fig. 4) is slightly lower than the energy of models using their source sub-family templates (ex: *Struct. 0 Seq. S1-2m0* and *Struct. 2 Seq. S1-0m2* on Fig. S7 and $M_{SQ}/Q$ on Fig. 4). This trend is true for all query/source pairs of sub-families and all strategies except for generated sequences using the fourth strategy (local background subtraction of query latent space using a KD-tree and the addition of source latent space), see Figs. S8, S9, S11, S13 and "Methods".

In this strategy, the modeled structures using generated sequences do not display energy distributions in-between the energy distributions of the original sequences modeled on structures of the query or of the source sub-families (dotted lines). The energy distribution of generated sequences modeled on structures belonging to the sub-family of their query latent space sub-family (ex: *Struct. 0 Seq.S1-0m2*, blue dots $M_{QS}/Q$) with the fourth strategy is closer to the energy distribution of the modeled structures using a sequence and a structure template from the same sub-families. The energy distribution of generated sequences modeled on structures corresponding to the sub-family of their source latent space (ex: *Struct. 2 Seq.S1-0m2*, orange dots $M_{SQ}/Q$) with the fourth strategy is closer to the energy distribution of the modeled structures using a sequence and a structure template from different sub-families. This indicates that the fourth strategy is less robust to latent space arithmetic than the other three strategies. No clear differences could be observed between the first, second, and third strategy.

## DISCUSSION

In this study, an Adversarial Autoencoder (AAE) architecture is proposed to analyze and explore the protein sequence space regarding functionality. Previous works based on Variational Autoencoder (VAE) have successfully reported the ability of this deep learning framework to model protein sequence and functional spaces (*Sinai et al., 2017*), predict amino acid fitness impact (*Hopf et al., 2017*; *Riesselman, Ingraham & Marks, 2018*), look into protein evolution (*Ding, Zou & Brooks, 2019*) or design new protein (*Greener, Moffat & Jones, 2018*). AAE networks have the advantage over VAEs to condition the latent space over a prior distribution which has previously been reported to have better efficiency (*Kadurin et al., 2017*).

Similarly to previous works using VAE architectures, this study has analyzed the capacity of the AAE architecture to correctly disentangle protein functional spaces of different families. The generative capacity of the models have been looked into with two original tasks: protein sequences interpolation between different sub-families and protein sequence arithmetics to mix properties of two sub-families.

AAEs have been trained on protein sequences of families known to have different sub-families with specific functions. The results have highlighted the ability of AAEs to separate sequences according to functional and taxonomic properties for the three studied families. This result emphasizes the ability of the AAEs to extract biological relevant features and encode them accordingly into a learned latent space.

Interpolations in the latent space between encoded sequences of different sub-families have shown smooth transitions of their amino acids even at the active site positions. The generated sequences along the interpolation paths can be considered as intermediate sequences with sequence properties linked to functions similar to their closest sub-families forming the start or end points of the path. The generated sequences have Shannon entropy values per amino acid position lower than biological sequences which indicates a lower amino acid diversity at each position. This trend point out the robustness and compactness of the latent space as the interpolation method takes the shortest path between the two points of the sub-families.

Finally, three strategies have been explored to generate protein sequences with features from two different sub-families. These strategies are based on latent space arithmetic, a concept previously applied in image generation tasks to produce relevant images with intermediate features (*Radford, Metz & Chintala, 2015*). Three out of the four different experiments carried out have been able to generate sequences with intermediate features as measured from their protein sequence similarity distributions and modeling energy assessments. Biological experiments will be needed to confirm the functional relevance of the transferred features, but the strategies could have many applications should it be validated.

The absence of measured differences between three out of four strategies used to generate intermediate sequences may also indicate that more optimal approaches could be designed. Similarly, the model architecture could also be improved. Currently the model input is a filtered MSA. An improved model could make use of full protein sequences of different sizes without filtering. Unfiltered protein sequences may benefit the generative model by capturing during training important protein specific motifs for family sub-functions (*Das, Dawson & Orengo, 2015*) not reaching the filtering thresholds.

Recent advances have been made regarding the protein sequence universe representation notably using self-supervised approaches, notably with the Transformer architecture (*Alley et al., 2019*; *Heinzinger et al., 2019*; *Rao et al., 2019*; *Rives et al., 2019*; *Strodthoff et al., 2020*). New models from image synthesis could also provide interesting approaches for the generation of protein sequences (*Vahdat, Kreis & Kautz, 2021*). The reported techniques in this study can be applied to any latent space projection and it would be interesting to combine them with representation of the protein sequence universe to navigate and perform feature transfer between protein families. These techniques could perhaps lead to the rediscovery of evolutionary sequence paths leading to the current protein families (*Alva et al., 2010*), improving our understanding of the protein sequence universe (*Dryden, Thomson & White, 2008*).

## CONCLUSION

This study shows that AAE models are able to finely capture the protein functional space of three different protein families with known sub-functions. The presented experiments carried out on the sulfatase family provide new insight on the effectiveness of generative model and protein sequence embedding to study and model protein function and

evolution. The proposed methods are robust to artifacts and generate consistent sequences and structures.

The results of this study show that AAE, in particular, and deep learning generative models in general, can provide original and promising avenues for protein design and functional exploration.

## GLOSSARY

The following glossary defines the different terms and techniques used in this manuscript.

| | |
|---|---|
| **Adversarial Auto Encoder** | A neural network architecture used for generative tasks. The architecture combine an auto-encoder and a generative adversarial network. |
| **Encoder** | Part of the AAE used to project the input data to a latent space. |
| **Latent space** | Input data point representation in a lower dimension. |
| **Decoder** | Part of the AAE used to reconstruct the input data from the latent space. |
| **Query sub-family** | Starting sub-family in interpolation experiments. Sub-family whose individual sequence latent spaces have been used in combination with the mean latent space of sequences from a source sub-family in latent space arithmetic strategies. |
| **Source sub-family** | Sub-family whose mean latent space has been used in combination with individual sequence latent space of a query sub-family in latent space arithmetic strategies. |
| **Target sub-family** | Sub-family used as end point in interpolation experiments. |

## ACKNOWLEDGEMENTS

TBF thanks the NVIDIA society for providing a TitanXp GPU to perform computations.

### Funding

This work was financially supported by ISCD Sorbonne Université, Paris, France. The funders had no role in study design, data collection and analysis, decision to publish, or preparation of the manuscript.

### Grant Disclosures

The following grant information was disclosed by the authors:
ISCD Sorbonne Université, Paris, France.

### Competing Interests

The author declares that they have no competing interests.

## Author Contributions

- Tristan Bitard-Feildel conceived and designed the experiments, performed the experiments, analyzed the data, performed the computation work, prepared figures and/or tables, authored or reviewed drafts of the paper, and approved the final draft.

## Data Availability

The code is available at GitHub: https://github.com/T-B-F/aae4seq.

## Supplemental Information

Supplemental information for this article can be found online at http://dx.doi.org/10.7717/peerj-cs.684#supplemental-information.

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
