# Peer review of "Navigating the amino acid sequence space between functional proteins using a deep learning framework"

_PeerJ Computer Science, doi:10.7717/peerj-cs.684_

## Round 0.1 · original submission · Minor Revisions

Two reviewers both gave minor correction recommendations. As a result, the paper is recommended as minor correction.

·

Basic reporting

A proofread for language clarity and minor errors is advisable (especially in the abstract)
Intro and background cover the majority of necessary topics
The structure is clear and figures are close to ready with minor tweaks
Raw data and code all seem to be openly available. I'd like the naming system for the data files to be made clearer in a metadata file or in the supplementary PDF though.

Experimental design

I very much like the application of AAEs to this topic. The exploratory and descriptive aspects of the work are pretty well defined. However the conditions for success in the design aspects are less clearly articulated (see below). The inclusion of the necessary code greatly aids replicability.

Validity of the findings

The author will need to be more cautious in their ever-interpretation of several results. Specifically, the protein designs were not experimentally validates, so there is no guarantee that they have the predicted function. Even seemingly minor differences to native sequences can have large effects on activities. In the absence of a collaboration to biochemically characterise the designs (likely beyond the scope of this work), care should be taken to be clear that the method has generated potentially designs (suggested to be promising by several in silico test). It is noted in L485-488, but needs to be more consistently framed throughout e.g. clarify "successfully transfer" versus "predicted to successfully transfer" or "likely to transfer".

Additional comments

Abstract & intro

L14 - "Protein sequence / function space" be cautious to be clear whether you're referring to sequence space, function space, or the space of mappings from sequence-to-function (which can be significantly different in both concept and structure).

L16 - The paper doesn't really end up differentiating between "relationships between protein positions and functions" versus "sequence patterns associated with functions". Might be better to combine these into a single concept for the abstract.

L21 - State the three model families here!

L23-25 - "The study also reports and analyzes for the first time two sampling strategies based on latent space interpolation and latent space arithmetic to generate intermediate protein sequences sharing sequential and functional properties of original sequences issued from different families and functions." Be very careful with this statement. Yes, the strategies generate intermediate protein sequences. No, they don't necessarily share functional properties (would need to be experimentally verified via recombinantly expressed protein)! Handled better at L351's "correspond to plausible sequences".

L30 - Again "successfully transfer functional properties between sub-families". Not experimentally confirmed. These protein designs could turn out to be non-functional (e.g. see success rates for Hilvert's or Baker's designs e.g "Robust design and optimization of retroaldol enzymes" doi.org/10.1002/pro.2059)

Methods

L148 - For the initial deep learning model, were amino acids each treated as different characters, or biochemical similarities taken into account? Note: there are multiple ways to encode protein sequence information for (e.g. "Evolution of Sequence-Diverse Disordered Regions in a Protein Family: Order within the Chaos" doi.org/10.1093/molbev/msaa096, "A method to predict functional residues in proteins" doi.org/10.1038/nsb0295-171, "Principal components analysis of protein sequence clusters" doi.org/10.1007/s10969-014-9173-2).

L176 - What was the dimensionality before dimensionality reduction?

Results

L252 - It would be worth noting why gappy columns were removed in this instance. I.e. In what way do they otherwise skew results?

L252 - How were remaining gaps handled? Were they simply treated as another character as with other amino acids, or encoded differently?

L272 - some of the colours of Fig 1 are extremely similar, making the in-image legend hard to follow. I recommend in at least panel A, overlaying the labels 1 to 11 over / next to their respective clusters

L272 - It's worth noting Fig S3 equivalent for HUP and TPP in the legend of Fig 1A.

L338 - some text on Fig 2 is unreadably small. I recommend enlarging the text on the right of panel B, or possibly making the panels vertically arranged so that each is full page width

L386 - "(Panels A and D)", I assume of Fig S6?

Discussion & conclusion

L482 - The comparison to image generation (or other AAN applications) might be interesting to note in more detail. How do these tasks compare in practice (e.g. accuracy, separation of clusters)?

L469-487 - How do the observations here compare to VAEs or similar methods? E.g. is the smooth transition during interpolation unique to this method or a common feature?

L508 - "Promising avenues" rather than "solutions" probably more accurate (assuming solutions in the sense of solution to a problem, rather than solution to an equation)


Minor language notes

A proofread for clarity and minor errors is advisable (especially in the abstract)
L16 - Missing comma between "functions, capture"
L17 - Probably best to include an oxford comma between "functions, or" to be safe
L17 - probably plural areas
Some sentences also get quite long. I recommend a quick read of the Structure of Prose and Stress Position sections of www.americanscientist.org/blog/the-long-view/the-science-of-scientific-writing.

Reviewer 2 ·

Basic reporting

1. The most important issue is, that the difference between source, queryand target families is not clear. In the methods section, source and query sub-families are nicely explained (l 213 - 214). However target sub-families are first mentioned in the results section (l 323). It seems like the source - query pair is used for latent space arithmetic and query - target pair for interpolation, but this is never clarified. For the reader to understand it better either use the same word pair throughout or clearly explain the different word pairs and why they should be distinguished.

2. Another important point is the improvement of language. For example, a few sentences that should be inspected are in the following lines:
l 16-20, l 28-31, l 56-58, l 465-467, l 477-479
Other small grammatical errors like missing words or conjugation of verbs should be corrected in the following lines:
l 41, l 50, l 56, l 110, l 188, l 195, l 381, l 438, l 461-462, l 492, l 493, l 495

3. Another important issue that needs clarification concerns figure 3 including corresponding caption and flow text (l 389 - 409). The caption of the figure is clear on its own, but more difficult to understand when looking at the flow text. There seem to be two paragraphs describing the same figure in different words. Lines 401 - 409 correspond to the figure captions and the sub-families given in the caption are very helpful. However, the bullet points (l 392 - 396) are difficult to relate to the description of the symbols in figure 3. As mentioned under issue 1, there is a confusion between the source - query and query - target pairs.

4. One less important point is a missing reference for latent space arithmetic (l 209 - 210, l 482 - 483). One refernence is mentioned in the results section (l 358), but to make this clearer you should also refer to it in the methods and discussion.

5. Another point concerns the Shannon entropy. It nicely visualises the variability in biological protein space compared to the generated sequences. But the Shannon entropy is first mentioned in the results (l 329). It is not entirely clear how and why it is used. To clarify this, you could describe it to the methods and add a reference (see experimental design).

Experimental design

1. The methods are generally well referenced and described, except for the Shannon entropy calculation, as mentioned above. Please add an explanation and reference (e.g. Jost, 2006, Oikos, 113) to the methods.

2. The names of the scripts are very descriptive and show their purpose. However, you could make the scripts more useful for other researchers by including a README to the scripts folder.

Validity of the findings

1. Statistical measures would make the findings more valid, i.e.R2 for fitted curves or lines in figure 2. You could also add if there is a statistically significant difference between the groups in figure 3.

Additional comments

1. A glossary could be of help, especially for the issue addressed above (1.Basic reporting). This would avoid confusion, for example between the source, query and target sub-families.

2. I had a look at the older version of the paper (”Exploring protein se-quence and functional spaces using adversarial autoencoder”, 2020) containing more figures in the main text. I would support the decision to move some figures from the supplementary in the current version back to the main text. Especially supplementary figure 2 could be helpful in the mainmanuscript.

3. You could elaborate more on how much of the sequence space is known. Useful references are for example Taverna and Goldstein (2002, Proteins, 46), Goldstein and Pollock (2016, Protein Science, 25), Marchi et al. (2019, PLOS Computational Biology, 15) or Alva et al. (2010, Protein Science, 19).

---

## Round 0.2 · accepted · Accept

The paper has been improved and revised following the comments from the reviewer. As a result, the paper is ready to be accepted.